

# MESMER v1.0.0: Consolidating the Modular Earth System Model Emulator into a Sustainable Research Software Package

Victoria M. Bauer[1], Mathias Hauser[1], Yann Quilcaille[1], Sarah Schöngart[1], Lukas Gudmundsson[1], and
Sonia I. Seneviratne[1]

[1]Institute for Atmospheric and Climate Science, Department of Environmental Systems Science, ETH Zurich, Zurich,
Switzerland

**Correspondence:** Victoria M. Bauer (victoria.bauer@env.ethz.ch)

**Abstract.** We present version v1.0.0 of MESMER – the Modular Earth System Model Emulator with spatially Resolved output. MESMER is a comprehensive software package for spatially resolved climate emulation. Version 1.0.0 revises the source code of several emulation approaches developed as part of earlier versions of MESMER and integrates them into a single python package that is readily available to the scientific community. We present the different MESMER components and the
development strategy used to integrate them. MESMER v1.0.0 is numerically stable, faster than previous versions, employs a modern data structure, and better disentangles data handling and the statistical core. We demonstrate that MESMER v1.0.0 is now a sustainable research software according to requirements for modularity, code quality, documentation, and reproducibility. Available output variables are annual and monthly mean temperature as well as several climate extreme indicators. The calibration of MESMER can take from a few seconds up to half an hour, while large ensembles of climate realizations can
now be produced (emulated) in just a few minutes. We facilitate the adoption of MESMER for the broad research community by providing pre-calibrated parameters, in a first step, for the emulation of annual mean temperature. MESMER v1.0.0 represents a crucial milestone in the coordination of on-going and upcoming developments of the MESMER Earth System Model emulator.

## 1 Introduction

Climate model emulators are statistical models trained to reproduce—emulate—the output of Earth System Models (ESMs). Such emulators reproduce the statistical properties of a reduced set of climate variables through training on ESM output. This makes emulators more computationally efficient than ESMs, which explicitly resolve the physical processes that govern the Earth system. While simple climate models emulate only the response of the Earth system at a global scale (Nicholls et al., 2020, 2021), spatially-resolved climate model emulators approximate the local response of ESMs (Beusch et al., 2020b;
Tebaldi et al., 2022; Womack et al., 2025; Mathison et al., 2025). Some of these emulators also have the capacity to represent natural variability on a regional scale (e.g. Beusch et al., 2020b) and can thus be used to produce large ensembles of realizations consistent with ESMs at a much smaller computational cost.



Their reduced complexity and low computational cost makes spatially-resolved climate model emulators a powerful tool for assessing a wide range of future scenarios or counterfactual climate trajectories. As such, spatially-resolved climate model emulators have recently been applied in a variety of research fields such as exploring overshoot scenarios (Schwaab et al., 2024), the modelling of natural variability (Nath et al., 2024), the attribution of climate extremes to emissions from countries (Beusch et al., 2022a) and income groups (Schöngart et al., 2025), and the integration of local observations in climate projections (Beusch et al., 2020b). Given this large range of applications, spatially-resolved climate model emulators matter not only for the exploration of scenarios and the linking to other modeling frameworks, but are also powerful tools for e.g., climate impact assessments and policy design. Consequently, there is a need to make such spatially-resolved emulators accessible to a broad community of researchers and to provide detailed documentation on how to use them.

Here we present MESMER v1.0.0 (hereafter called MESMER v1), a comprehensive software package for spatially-resolved climate model emulation. MESMER is a climate model emulator that translates annual mean climate trajectories into spatial fields of selected climate variables over land, based on training data from ESM simulations. It accounts for the local forced climate response as well as internal variability of the respective ESM and produces spatio-temporally coherent realizations. Since its original development by Beusch et al. (2020b), several components have been added to MESMER to emulate more variables and temporal resolutions, resulting in an organic growth of the code base (Beusch et al., 2020b, a; Nath et al., 2022; Quilcaille et al., 2022, 2023; Schöngart et al., 2024).

In MESMER v1, multiple former components are integrated into a single package. This new and integrated version of MESMER comes with a complete redesign of the Application Programming Interface (API) to make the source code more modular and user-friendly. We implement a new data structure that simplifies the data flow within MESMER as well as the loading and saving of data. Furthermore, we update parts of the previously published methodologies to improve the stability of MESMER output and reduce its run time. This work presents the first integral release of MESMER and also constitutes a guide to future users and contributors.

## 2 MESMER: past developments

### 2.1 Overview

The initial version of MESMER was developed to generate large ensembles of spatially resolved land surface annual mean temperature time series that mimic ESM output (Beusch et al., 2020b, a). Subsequently, the MESMER ecosystem grew, adding more capabilities: MESMER-M extends the emulator to monthly local temperatures (Nath et al., 2022). Similarly, MESMER-M-TP emulates monthly local temperature and precipitation (Schöngart et al., 2024). Lastly, MESMER-X introduced capabilities to emulate various annual indicators of climate extremes and climate impact drivers, such as the maximum annual temperature (Quilcaille et al., 2022) or soil moisture (Quilcaille et al., 2023). Despite using different statistical methods, each approach separates the local signal into a *local response* and *local variability*, providing a common ground for the design of MESMER v1. In the next sections we will shortly discuss the (statistical) methods used for each of these MESMER components.



## 2.2 Annual temperature (MESMER)

The emulation of spatially resolved land surface annual mean temperatures consists of two main modules: (1) the local response module, which translates the global mean temperature into the local temperature response on a gridcell level (Tebaldi and Arblaster, 2014; Herger et al., 2015; Seneviratne et al., 2016; Wartenburger et al., 2017) using linear regression. (2) The local residual variability module generates temporally and spatially correlated random samples, with an order 1 auto-regressive (AR(1)) process and spatially correlated innovations. To dampen spurious correlations from rank deficient empirical covariance matrices, a Gaspari-Cohn localization is employed (Cressie and Wikle, 2011; Humphrey and Gudmundsson, 2019) (see Table 1, and Beusch et al. (2020b)). We refer to the state of the MESMER code base before the rewrite as the original MESMER version (MESMER v0.8.3 in Hauser et al., 2021), corresponding to the release for Beusch et al. (2022b).

**Table 1.** Overview of MESMER components. Each component estimates a local response and the local residual variability. AR(1) stands for auto-regressive process of order 1, PCA, stands for principal component analysis.

| Component | Local response | | Local variability | |
|---|---|---|---|---|
| | Description | Statistical Methods | Description | Statistical Methods |
| **MESMER** | Estimate local response to global mean temperature | Linear regression | Generate local temperature variability | AR(1) process with spatially correlated innovations, Gaspari-Cohn localization |
| **MESMER-M** | Generate monthly seasonal cycle around annual mean temperature | Harmonic model | Transform skewed local residuals to more closely follow a normal distribution | Yeo-Johnson Power Transformer |
| | | | Generate local monthly temperature variability | Cyclo-stationary AR(1) process with spatially correlated innovations, Gaspari-Cohn localization |
| **MESMER-M-TP** | Generate precipitation component that co-varies with temperature | PCA, Generalized Linear Model | Generate spatially-coherent residual precipitation variability | PCA, Kernel Density Estimation |
| **MESMER-X** | Estimate parameters of the distribution conditioned on a function of a global mean climate variable | Conditional distribution | Generate realizations of the desired climate variable and transform to the target distribution | AR(1) process with spatially correlated innovations, Gaspari-Cohn localization, Probability Integral Transform |





## 2.3 Monthly temperature (MESMER-M)

MESMER-M is the **m**onthly emulation component of MESMER. It takes in gridded local annual mean temperature and emulates gridded local monthly temperature. MESMER-M relates the annual mean temperature at each gridcell to its monthly temperature response using a harmonic model. To make potentially skewed residuals more normal a Yeo-Yohnson Power Transformer (Yeo and Johnson, 2000) is employed. The residual variability module emulates spatially and temporally correlated noise for each month, employing a cyclo-stationary AR(1) process (Cressie and Wikle, 2011). The the same approach is used to localize the empirical covariance matrix as in the original MESMER. The details of the method are presented in Nath et al. (2022) and outlined in Table 1.

## 2.4 Monthly precipitation and temperature (MESMER-M-TP)

MESMER-M-TP is a monthly **p**recipitation extension to MESMER-M (Schöngart et al., 2024). It generates gridded monthly mean precipitation fields using gridded monthly mean temperature fields as input. The emulator is designed to realistically capture the covariance structure between temperature and precipitation, ensuring that the generated precipitation fields are spatially and temporally consistent with the input temperature fields. Precipitation at each gridcell and for each individual month is modeled as the product of two components: The first component describes how precipitation statistically co-varies with temperature, modeled with a Generalized Linear Model, specifically with a gamma distribution with a logarithmic link. Here, temperatures from neighboring gridcells (within a configurable radius) serve as predictors for local precipitation responses. The second component accounts for variability in precipitation not explained by temperature. While assumed temperature-independent, this residual term can still reflect spatial correlations across the precipitation field. To capture this structure, the residuals are transformed via principal component analysis (PCA), and their distribution is estimated using kernel density estimation with a Gaussian kernel for resampling (Tab. 1).

We note that MESMER-M-TP is not fully integrated in MESMER v1, but we include it here for completeness. The full integration of MESMER-M-TP is expected to follow in the next MESMER release.

## 2.5 Annual extremes and climate impact drivers (MESMER-X)

MESMER-X was developed to emulate local climate e**x**tremes and climate impact drivers (Quilcaille et al., 2022). Instead of a local response module as employed in MESMER it uses non-stationary distributions, i.e. distributions of the climate variable that are conditional on covariates, e.g., the global mean temperature. Therefore it can accommodate non-normal distributions and non-linear responses to changes in global climate. This module is designed to be applicable for a large range of variables, thus allows various distributions (Normal, Generalized Extreme Value, etc.) and covariate responses (linear, polynomial, sigmoid, etc.). The generation of realizations is achieved by sampling an AR(1) process as in MESMER, and subsequently using Probability Integral Transforms (Angus, 1994; Gneiting et al., 2007) to transform the drawn samples to the distributions of the climate indicators (Tab. 1).





## 3 Design of MESMER v1

The development of MESMER and its extensions took part over several years and was guided by scientific interest and relevance. The code was developed by multiple researchers, to a large extent independently of one another, and without a focus on software design. Due to its relevance to the community and use in various international projects, a rewrite, integration of all extensions, and unification of the code base was deemed necessary. In doing so, we followed requirements for sustainable research software, which are that the software is maintainable, extensible, testable, accessible, flexible, has a defined software architecture and is documented (Loewe et al., 2021).

As shown above, each MESMER component has a unique structure and uses different methods to achieve its objective. However, each of the components is calibrated on ESM output to obtain parameters specific to each ESM. Further, each component emulates a *local response* to their respective predictor variable(s) and then generates *local variability*. Fig. 1 shows the synergies between the existing MESMER components, e.g., all use the same method for localizing the empirical covariance matrix for the non-parametric sampling of local variability. This perspective opens possibilities for future developments. Additions to MESMER need not be limited to one component or span all of them but can fall into any of the domains defined here.

### 3.1 The modular user interface

MESMER v1 is a pure python package and contains statistical functionality and utilities to process ESM data to be used in the statistical functions. The statistical functions and classes are organized in two modules: `mesmer.stats`, and `mesmer.proba` (Fig. 2). The module `mesmer.proba` holds classes and functions related to conditional probability distributions, i.e., functionality that is mainly used in MESMER-X such as fitting a conditional distribution (`mesmer.proba.ConditionalDistribtion().fit`) or the probability integral transform (`mesmer.proba.ProbabilityIntegralTransform`). The module `mesmer.stats` contains all other statistical classes and functions, including linear regression (`mesmer.stats.LinearRegression`), auto-regression (e.g., `mesmer.stats.select_ar_order`), and others. The data handling functionality is distributed over several modules, and includes functions to compute anomalies (`mesmer.anomaly.calc_anomaly`), the global mean (`mesmer.weighted.global_mean`), or for pooling data of several scenarios and ensemble members of ESM runs (`mesmer.datatree.pool_scen_ens`).

This layout leads to the following new workflow for the user: for *calibrating* MESMER they need to (1) load the predictor data, (2) calibrate the parameters using the statistical functionalities and (3) save the parameters. To *create emulations*, users need to (1) load the ESM-specific parameters again, (2) calculate the (deterministic) local response, (3) draw samples of the local variability, and (4) save the emulations. This restructuring is more in line with the framework shown in Fig. 1. It allows users to interact with the statistical methods directly, and every step of the computation is clearly separated. It disentangles the data handling from the statistical functionality, and the data loading and saving, and organizes the code into small units only concerned with one well-defined task, making the code more modular (Trisovic et al., 2022).



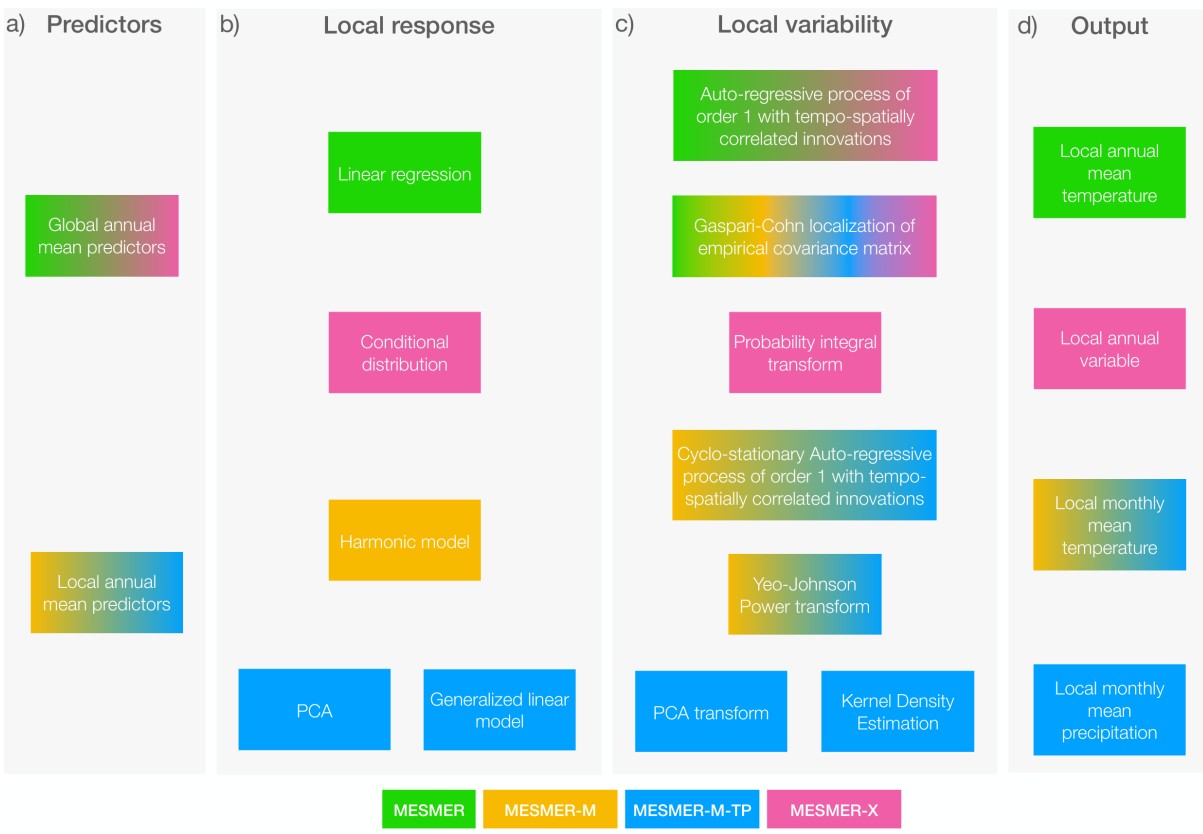

**Figure 1.** Synergies between previously published MESMER components (indicated by color) showing predictors (a), methods for the local response (b), local variability (c), and output variables (d) that are used within MESMER.

## 3.2 Data Structure

MESMER v1 uses a new data structure based on *xarray* (Hoyer and Hamman, 2017). *xarray* provides multi-dimensional labeled arrays and datasets in Python. It adopts Unidata's self-describing Common Data Model on which the network Common Data Form (netCDF) (Rew and Davis, 1990) is built. One main advantage over the previously-used *numpy* array manipulation library (Harris et al., 2020) is the introduction of labeled array dimensions, as well as providing coordinates to these named dimensions and adding attributes to arrays and datasets. This facilitates keeping track of multiple dimensions present in the climate data used in MESMER, e.g. latitude, longitude, time, or initial condition member. Several xarray arrays (`xarray. DataArray`) can be collected into `Dataset` objects in the form of named data variables if their coordinates align, e.g. they live on the same grid and have the same time steps. Several datasets in turn can be held in a so called `DataTree`, which allows for a hierarchical storage of data that lives on different coordinate systems.

MESMER was designed to be calibrated on ESM model output from the Coupled Model Intercomparison Projects (CMIP), e.g. CMIP6 (Eyring et al., 2016). We combine historical simulations with future projections, such as the shared socioeconomic



**Figure 2.** Public API of MESMER v1. Light yellow boxes indicate functions, blue boxes classes, and lilac boxes class methods. Gray frames mark individual python files, i.e. (sub-)modules.





pathways (SSPs), (O'Neill et al., 2016). Most models provide several scenarios, and historical and future data do not have the same time coordinates. Therefore, different scenarios are stored in `DataTree` objects in MESMER. Predictor data for, e.g. global mean temperature (`tas`) and optionally squared global mean temperature (`tas2`) or global mean ocean heat content (`hfds`), are held in Datasets at each node of the tree (Fig. 3a). Similarly, the target variable for each scenario, e.g., gridded near surface temperature, is stored in a separate `DataTree` object (Fig. 3b). Each variable in the datasets is a `DataArray`

with a time dimension, and for the gridded data, an additional gridcell dimension. If several initial condition members are available for the same scenario, these can be stored along a member dimension. It is thereby important, that the predictor and target trees are isomorphic, i.e. they hold the same scenarios, and that in each scenario, the coordinates align, so that each target sample also has a corresponding sample in the predictor data. For several of the statistical routines, the data must be pooled into a single sample dimension, where the scenarios are a new dimension, and the scenario, time, space and member dimensions

are stacked into one (Fig. 3c & d). MESMER provides a routine for the pooling, ensuring that no missing values occur from incorrect stacking and predictor and target data are aligned (`mesmer.datatree.broadcast_and_pool_scen_ens`).

### 3.3 Persisting calibrated parameters and emulations

The adoption of *xarray* data structures allowed us to effortlessly save parameters as *netCDF* files, instead of the previously used *pickle* format. *netCDF* is a widely used scientific data format and has several advantages over *pickle* files in that it saves

independent of the underlying python structures, which makes it more secure, interoperable with most common programming languages and ensures it can seamlessly be read in the future.

Together with MESMER v1 we publish pre-calibrated parameters for the emulation of annual mean surface temperature following the approach of the original MESMER (Beusch et al., 2020b) on Zenodo (Bauer et al., 2025). These are calibrated for 58 CMIP6 models (Eyring et al., 2016) post-processed according to Brunner et al. (2020), using all available initial condition

members of SSP1-1.9, SSP1-2.6, SSP2-4.5, SSP3-7.0 and SSP5-8.5 for each model. We calibrate using only global mean surface temperature as a predictor. This set of parameters allows users who are only interested in emulations of annual mean surface temperature to omit the calibration step. We thereby establish a consistent parameter-set for future applications. This parameter-set will be maintained together with MESMER and can be extended depending on the interest of the research community.

### 3.4 Documentation

To ensure that MESMER v1 is accessible to a broad range of users, we considerably extended the documentation of the software. We created a MESMER documentation webpage (https://mesmer-emulator.readthedocs.io/, Fig. 4a), which is automatically generated for each version of the code to ensure it is always up-to-date. Highlights of the new documentation webpage include the API reference, which consists of the rendered in-code documentation of all public functions and classes

(Fig. 4b), and multiple tutorials that demonstrate the workflow of the most important MESMER use-cases, therein calibrating a real (but coarse resolution) example ESM and creating emulations for the given ESM, for the MESMER, MESMER-M and MESMER-X workflow (Fig. 4c). These tutorials are rendered from interactive jupyter Notebooks, also available from the



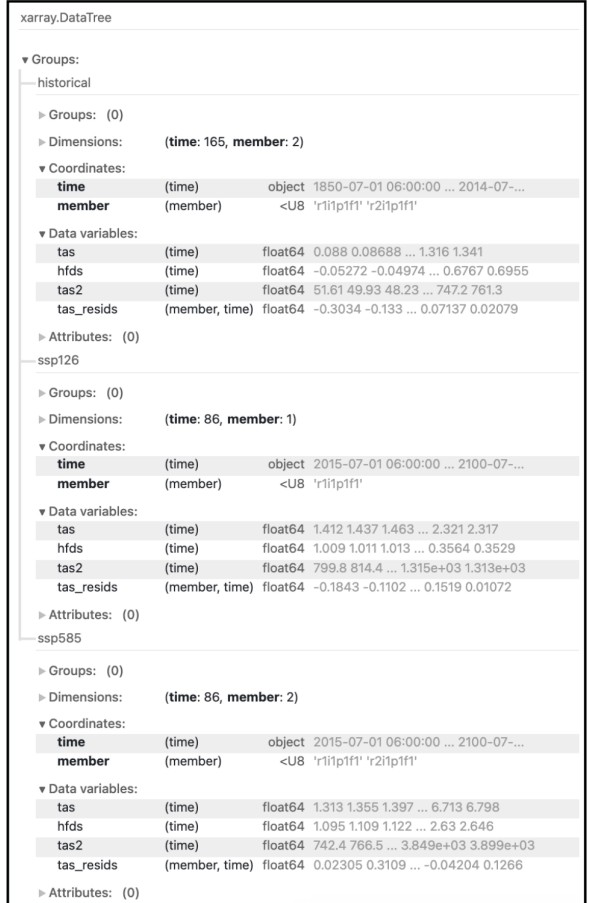

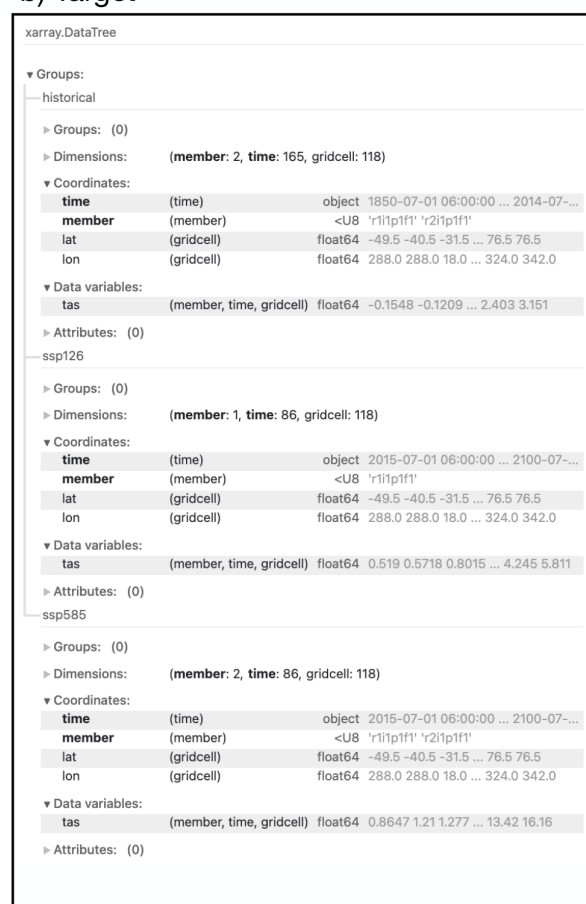

**Figure 3.** Data structure in MESMER. DataTrees holding predictor (a), and target data (b), respectively, where each scenario is in a separate node. Predictor (c), and target data (d) stacked into Datasets along a sample dimension, respectively.




MESMER repository (see Figure caption for link) for users to download. Moreover, the documentation also contains the open software license of MESMER, notably the GNU General Public License as published by the Free Software Foundation, version

3 (cf. https://www.gnu.org/licenses/gpl-3.0.html, last accessed: September 10th 2025) and resources for developers.

In addition, the MESMER github page (https://github.com/MESMER-group/mesmer) provides development oriented documentation. Here, the development team records all changes made to the software, discusses its design, collects issues, and structures the development workflow. In addition, this page is an access point for users to post issues or calls for improvement.

## 3.5 Testing

Re-structuring the MESMER code into small units allowed us to write extensive unit tests, which ensure each individual function and method works as expected. We make use of two kinds of tests in MESMER v1: unit tests and integration tests. Unit tests are used to test the smallest units of code in a program (i.e. a single function or method), while integration tests ensure that these units work as intended when combined (Wilson et al., 2014). The test suite of MESMER is automatically run every time a contribution is made to the MESMER code base on github, and can of course also be run locally by the developer.

The implemented unit tests include testing that errors are raised when expected, return values are of the expected type or shape, and class attributes are assigned as expected. More sophisticated unit tests test that an API fulfills its intended purpose. Such tests in MESMER v1 are for example if the fitting routine of the auto-regressive process or the linear regression can retrieve the correct parameters from simulated data with known parameters, or if the Yeo-Johnson Power Transformer actually reduces the skew of artificially skewed data.

While each MESMER component (MESMER, MESMER-M, MESMER-X) showed good performance in emulating the statistical properties of ESM output (Beusch et al., 2020b; Nath et al., 2022; Quilcaille et al., 2022, 2023), no integration tests were available for MESMER-M and MESMER-X. Therefore, the integration tests of MESMER v1 execute the whole calibration and emulation workflow for each MESMER component. Specifically, each component is calibrated using CMIP6 data that was re-gridded to a coarse resolution of 20x20 gridcells. Subsequently, one to ten realizations are drawn using

the calibrated parameters. The calibrated parameters and emulations are compared to previously-saved output. This ensures stability of MESMER functionality as well as the numeric stability over time, or between different machines. The latter is especially important for software that is used in a scientific context to ensure reproducibility (Reinecke et al., 2022). If a change is intended, the developer can update the saved test data.

## 3.6 Assessment of the redesigned Code base

We assess if MESMER v1 reaches the development goals (1) modular and comprehensible source code, (2) modern data structure, (3) internal and external documentation, (4) test coverage of the whole project, and (5) strategy for version control and continuous integration. We exclude tests, tutorials, documentation source files, and deprecated modules from the assessment, as well as the testing module, which is only thought for use in the tests.

**Modular and comprehensible source code.** Code is modular if one unit of code (e.g. a function, method or class) is only
205 dedicated to one task (Trisovic et al., 2022; Nyenah et al., 2024). Thus, it is reasonable to assume that modular code units





## a) Landing page

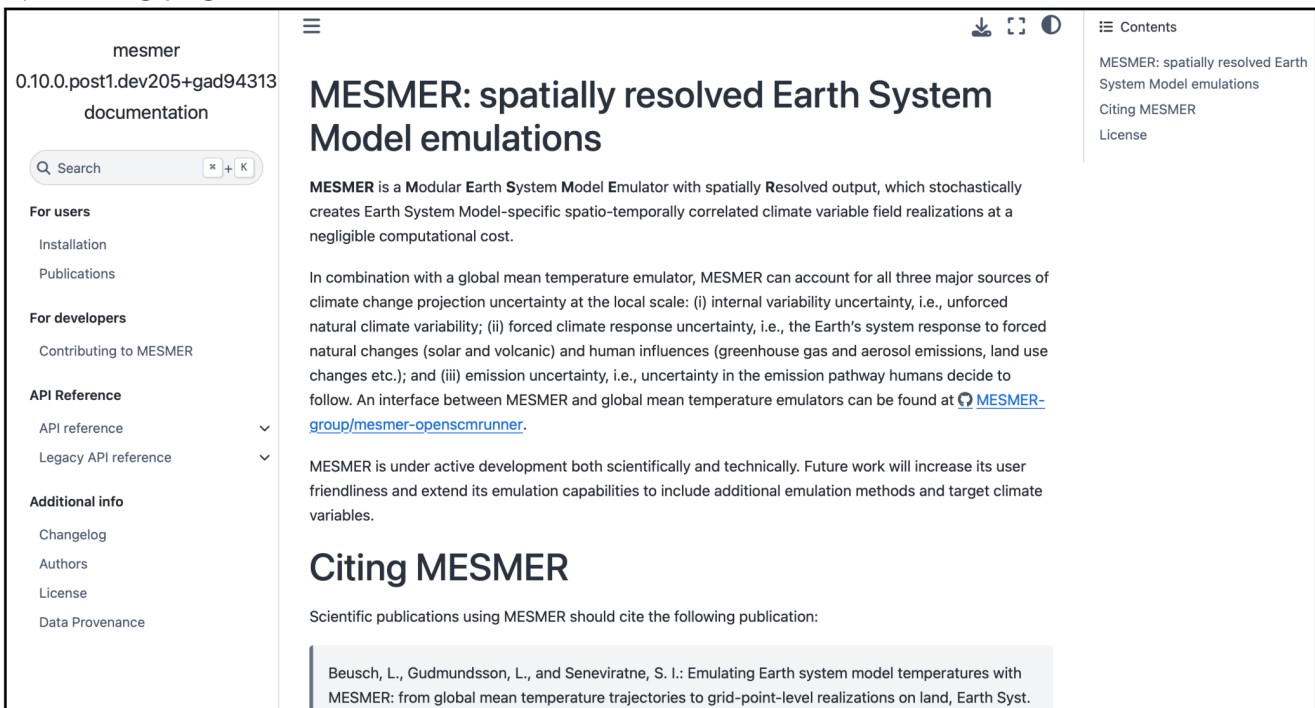

## b) API reference

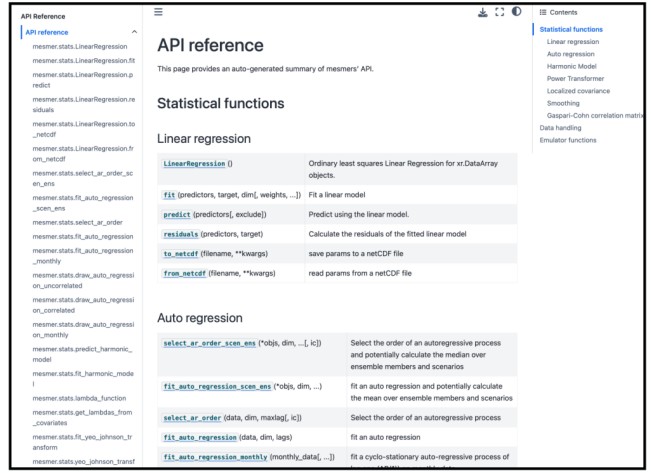

## c) Tutorials

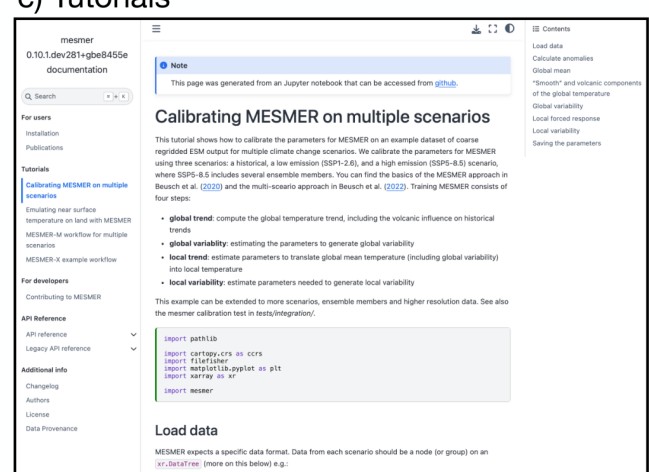

**Figure 4.** Screenshots of html documentation of MESMER: the landing page (a, https://mesmer-emulator.readthedocs.io/en/latest/index.html), API documentation (b, https://mesmer-emulator.readthedocs.io/en/latest/api.html) and one of several tutorials (c, https://mesmer-emulator.readthedocs.io/en/latest/tutorials/tutorial_mesmer_calibrate_multiple_scenarios.html).



are short. On average, source code files in MESMER v1 have 269 lines of code (Table 2). Lines of code here are lines of source code, including blank lines and comments, counted by the scc tool available from github (https://github.com/boyter/scc, last accessed: September 10th 2025). Nyenah et al. (2024) argue that code files shorter than 1000 lines can be considered modular, as they are still readable and likely understandable in a reasonable amount of time and likely to only be focused on one specific task. The longest source code file in MESMER v1 has 1219 lines. While this file exceeds the recommendation the second largest file has 629 lines. In addition to lines of code per file, we also assessed number of lines per object definition, i.e. per defined function, class or method. In total, the MESMER v1 code base about 8.000 lines of code in total and 214 defined objects, making for an average of about 38 lines of code per definition (Table 2), which we argue is reasonably short to understand. Furthermore, we employ the pylint score, to analyze the adherence of the MESMER v1 code to coding standards (Goodger and Rossum, 2011), as a metric of how comprehensible the MESMER v1 source code is. The score lies between 0 and 10 with 10 being perfect compliance. A score above 6 is considered satisfying (Molnar et al., 2020; Nyenah et al., 2024). The refactored code base reaches a score of 8.74 (Table 2). Note that here we excluded the deprecated legacy modules that will be removed in the next version. The most common violations of the standards in MESMER are: (1) functions having more than 5 arguments, which is sometimes unavoidable, (2) missing module docstrings, as we do not document modules but only functions, classes and methods, and (3) left over TODO comments in the code, these are kept for future developments and not urgent (not shown).

**Modern data structure.** We implemented a new data structure based on xarray. This library is well maintained externally and independent of MESMER. It is interoperable with all MESMER dependencies and provides a safe and convenient data saving and loading infrastructure. Moreover, the xarray data structure greatly enhances readability of the MESMER v1 source code.

**Internal and external documentation**. MESMER v1 provides internal documentation through docstrings and in-code comments and external documentation in the forms of a ReadME, the documentation webpage, the github page, and this publication. MESMER v1 provides documentation for 100 % of its public API and 53 % of private definitions (Table 2). We count a function, class or method as documented if it has a docstring attached to its definition that is longer than 10 characters.

**Test coverage.** MESMER has and automatically runs a large testing suite for every code contribution and release. MESMER has a test coverage of 99 %, meaning that 1981 of 1991 of tracked source code lines (which also exclude deprecated modules) are covered by at least one test.

**Strategy for version control and continuous integration**. The MESMER v1 source code lives on github. We implemented several workflows to manage a sustainable continuous integration, like code review, issue tracking, automatic code checks, and an automatic testing suite.

Overall, we conclude that MESMER v1 complies with standards for sustainable research software (Loewe et al., 2021; Reinecke et al., 2022; Nyenah et al., 2024).





**Table 2.** Assessment of development targets.

| Target | Metric | Value |
| --- | --- | --- |
| Modularity | Average number of lines of code per object | 38 |
| | Number of lines of code per file (avg [min, max]) | 238 [7, 1219] |
| Code quality | Pylint score | 8.74 |
| Data structure | Numpy arrays to xarray data structure | Completed |
| Documentation | % of public API documented | 100 % |
| | % of private API documented | 53 % |
| Testing | % of lines covered by tests | 99 % |

## 4 Performance optimization and numerical stabilization

We implemented several measures to optimize and stabilize MESMER. The performance improvements presented in this
section were measured using data from one initial condition member of the SSP5-8.5 scenario together with its matching
historical member of the IPSL-CM6A-LR model of the CMIP6 new generation data archive (Eyring et al., 2016; Brunner
et al., 2020). In favor of reducing testing time, we re-gridded the dataset of the yearly and monthly mean and maximum
temperatures to a resolution of 20x20 gridcells.

### 4.1 Stabilization of covariance matrix decomposition

Previous versions of MESMER produced numerically different emulations on different machines, despite using the exact same
parameters. We traced this problem back to numerical instabilities in the inversion of the spatial covariance matrix and were
able to resolve it by changing the decomposition strategy of the spatial covariance matrix from a singular value decomposition
to the Cholesky method (Gentle, 2009). The decomposition is needed to sample tempo-spatially correlated variability at each
gridcell, i.e. the white noise term of the AR(1) process (Fig. 1c). While the decomposition using singular value decomposition
is not unique and depends on linear algebra routines independent of the python installation on a machine, Cholesky decom-
position is unique and produces stable results on different machines (see also https://numpy.org/doc/2.3/reference/random/
generated/numpy.random.Generator.multivariate_normal.html, last accessed: September 10th 2025). We find that Cholesky
decomposition is about 80-100 times faster for a 118x118 matrix than singular value decomposition. Consequently, using the
Cholesky method has the added benefit of speeding up the calibration (7.8 times faster) and emulations (1.7 times faster). The
255 calibration step sees a bigger improvement because the the covariance matrix needs to be transformed several times during the
cross-validation step (15 cross-validations for 7 radii with our test data).

Additionally, we want to note that large localization radii can lead to singular localizer matrices and are thus not suited for
mitigating spurious correlations in the empirical covariance matrix. This phenomenon starts occurring at localization radii of
about 11.000 km and larger. Such large localization radii are only rarely selected by the cross-validation, except when there are
260 a large number of samples, e.g. a large number of initial condition members, are available, in which case the rank deficiency of



the empirical covariance matrix is small. In such cases, the large localization radius must be kept sufficiently low to preserve positive definite covariance matrix, thus still mitigating spurious correlations.

## 4.2 Performance optimization of the harmonic model and Power Transformer

We were able to reduce the time needed for the fit of the harmonic model from about four minutes to below one second for our test dataset. First, we only fit the Fourier Series up to an order of 6 instead of 12 by default, since a Fourier series of order 6 is able to capture a series with 12 features. Second, we now only fit until the first local minimum of the Bayesian Information Criterion is reached when finding the optimal order, as the BIC was found to increase monotonically thereafter. Third, we pass the parameters from the previous order as first guess when fitting the parameters of the Fourier Series for each new order. This improves the performance since the parameters of the lower order terms only vary slightly when adding a new order to the series. We also optimized the generation of the Fourier Series by factoring out constant terms, e.g. the harmonic representation of the months.

Testing the Power Transformer showed that it did not reliably reduce the skew of monthly residuals. We stabilized the Power Transformer by expanding the bounds imposed on the optimization of the parameters of the logistic function used for conditioning the normalization parameter, and switched the optimization method to Nelder-Mead. Moreover, we set the first guess for the fitting to assume the data is already normally distributed, which has proven to be a good first guess for the monthly temperature residuals. In addition, there were problems with the numerical stability of the back-transformation using the Power Transformer, that arose from floating point errors in the formulation of the transform. Specifically, round-tripping certain values did not lead to the original data for small numbers. We were able to fix these issues by switching to more precise routines. With these adjustments, the time needed for the calibration of the Power Transformer was reduced from approximately 8 minutes to under 5 seconds.

## 4.3 Switch to cyclo-stationary AR(1) process for monthly emulations

The method employed to emulate local variability for monthly temperatures is different from the approach for annual temperatures (Nath et al., 2022; Beusch et al., 2020b). The residual temperature variability of June may depend differently on the one of May than the one of July does on June. Therefore, there are separate AR parameters for each calendar month, and the method used for monthly temperature emulations is a cyclo-stationary AR(1) process instead of a AR(1) process (Storch and Zwiers, 1999). In contrast to the AR(1) process, a cyclo-stationary AR(1) process can still be stationary with AR-coefficients larger than absolute 1, since the destabilizing effect of one coefficient (e.g. of one month) can be compensated by another (e.g. the one of the next month). We thus removed the bounds of [-1, 1] enforced on the slopes during the fitting routine of the monthly AR(1) process and found that slopes outside these bounds do appear in our parameters. Consequently, we are not able to adjust the covariance matrix with the AR slopes anymore as done for e.g. annual mean temperatures (Beusch et al., 2020b; Storch and Zwiers, 1999). To prevent the overestimation of the driving noise variance, we now fit the covariance matrix on the residuals of the cyclo-stationary AR(1) process and found that for generated data, this leads to a better representation of the variability than estimated by the AR(1) process (not shown).





## 4.4 First guess module for the Conditional Distribution

MESMER-X emulates climate variables by sampling their values from a calibrated conditional distribution. Thereby, every parameter of the conditional distribution, e.g. the location or the scale, can be a function of covariates, i.e. global predictors such as the change in global mean temperature. Both the type of distribution and the functions for the parameters are chosen by the user, based on statistical and physical arguments. Identifying the adequate distribution and functions require data exploration and validation (Quilcaille et al., 2022, 2023). Because MESMER-X is designed to be able to emulate a large range of variables, the module to train these conditional distributions is designed for flexibility and robustness of the training. To maximize the chances of finding the best solution, an algorithm finds a first guess, as close as possible to the final solution. This algorithm builds on the principles introduced in (Quilcaille et al., 2022, 2023), which uses an iterative process to fine tune the first guess for each gridcell:

1. First approximation for the location: using only the function of the location, global minimization of the square of the errors on the mean of the sample and the first derivatives with the predictors.

2. Fit of the location: using only the function of the location, local minimization of the square of the differences between the evolution of the location and the smoothed sample. It implicitly assumes that the location and the mean are close.

3. Fit of the scale: using only the function of the scale, local minimization of the square of the differences between the evolution of the scale and the empirical scale. This empirical scale is calculated as the square of the differences between the smoothed sample and the sample itself. It implicitly assumes that the scale and this empirical estimate are close.

4. Overall improvement: using the distribution and all functions for its parameters, local minimization of the negative log-likelihood. Given the three former steps, the location and scale are relatively good approximations, but the remaining parameters need to be estimated as well. The location and scale may change here, to adapt to the evolutions of the overall conditional distribution.

5. Validation of the first guess: several tests are conducted to verify necessary conditions or user-defined constraints. Namely, the parameters must be within bounds (e.g. the scale must be positive); the coefficients within their functions may have to respect other bounds; the whole sample must be within the support of the distribution; every point of the sample must have a minimum probability, set by default to $1 \cdot 10^{-9}$. If all tests are verified, this solution will be used as first guess.

6. Eventual improvement: if the former step has shown failure of a test, an additional local optimization is conducted, using the minimization of the negative log-likelihood at a cubic power. This approach gives more weights to outliers, thus avoids having parts of the sample that will cause the test on the minimum probability to fail.

The global optimizer is a Basinhopping algorithm, with 10 iterations and 100 intervals, with a seed set for reproducibility. The local optimizer is a Nelder-Mead algorithm, but the user may provide a second local optimizer, and the best result will





be used. When minimizing the negative log-likelihood, weights can be used for every point of the sample, e.g. uniform or based on the density of the predictor samples. The predictor samples are often not equally distributed over their whole range, for example the global mean temperature has more samples at lower than high values because there is a longer period of relatively constant temperatures in the historical period. Therefore, a weighting may help in providing equal performance for the fit over the whole range of the predictors. This weighting is based on the inverse of the density, calculated with a Gaussian kernel-density estimate.

The solution for the conditional distribution is finally obtained by a local minimization of the negative log-likelihood from this first guess, accounting for given weights. Every iteration during the minimization is checked with the same tests outlined in step 5 of the first guess. The optimizer is again Nelder-Mead, with the possibility to use a second one defined by the user. Overall, the first guess module enhances the chances of finding a good fit of the conditional distribution.

## 5 Performance assessment

To illustrate the use-case of MESMER being used on a personal machine instead of a cluster, we calibrate and emulate three use-cases of MESMER on a personal machine with an Apple M3 Pro Chip (Nov 2023) and 18 GB of memory. The three use-cases are: calibration and emulation of local annual mean surface temperature (MESMER), local monthly mean surface temperature (MESMER-M), and local annual maximum surface temperature (MESMER-X) using only annual mean temperature as predictor. For the calibration we use CMIP6 data of one ESM, specifically IPSL-CM6A-LR. We perform two calibrations: once on a single member and once a 24-member ensemble of five future scenarios and the historical period with four initial condition members each. The calibration including one member is performed on the historical and SSP5-8.5 scenarios, using the "r1i1p1f1" member, yielding 251 samples. On the used 2.5° x 2.5° grid and after masking the ocean and Antarctica, this makes for a total of 2652 gridcells. The ensemble includes the initial condition members "r1i1p1f1", "r2i1p1f1", "r3i1p1f1", and "r4i1p1f1" of the scenarios SSP1-1.9, SSP1-2.6, SSP2-4.5, SSP3-7.0, and SSP5-8.5 respectively, together with the matching historical members. Note that we use the historical members only once (not once per future scenario). This yields a total of 6 x 4 = 24 members with 86 timesteps for the future scenarios, and 165 timesteps for the historical members, respectively, a total of 2380 samples on 2652 gridcells. While we used the same members for all scenarios for convenience and to limit the number of samples, this is not necessary and MESMER can be run with an arbitrary number and combination of ensemble members or scenarios. Note that we only timed one individual execution for each use-case, and did not run on multiple machines or control for work load from other programs (email, browser, etc.). This analysis does not aim to be a thorough benchmarking of the software as each application of MESMER depends heavily the use-case. Instead, we want to give rough time indications for future users and point out which parts of the calibration and emulation routines control the overall computation time.

Calibrating parameters for annual mean surface temperature emulations, takes approximately half a minute for one member and 4 minutes for the 24-member ensemble (Tab. 3). In both cases, most of the time in the calibration is taken up by the fitting of the local AR(1) processes with spatially correlated innovations (>85%, Tab. A1). This is, on the one hand, the fitting of the AR(1) parameters, and on the other hand finding the optimal localization radius for the empirical covariance matrix.



Importantly, the time needed to fit the AR(1) process depends mostly on the number of supplied samples, while the time needed to find the optimal localization radius depends primarily on how many radii need to be tested until a local minimum is reached. Calibration on one member makes the empirical correlation matrix severely rank deficient since there are only 251 samples for 2652 gridcells. This leads to a low localization radius, corresponding to a "strong" localization, in this case 2'250 km. In contrast, calibrating on multiple members provides more samples and thus mitigates the rank deficiency, resulting in a higher localization radius, here 4'750 km. Since we tested radii from 1'000 to 10'000 km in steps of 250 km, the calibration on one member cross-validated 7 radii, while the calibration on the ensemble needed 17 cross-validations. Note that we use a number of 30 folds for each cross-validation step throughout this analysis. Subsequently, the time needed to find the localization radius can be greatly decreased, if the user supplies a suitable range of radii to test.

The calibration of monthly mean temperatures (MESMER-M) takes significantly longer: Calibrating on one member takes five minutes and calibrating the the ensemble takes more than 25 minutes. Again, most time (about 80 %, or 20 minutes for the ensemble) is spent on finding the optimal localization for the empirical covariance matrices (Tab. A2). In contrast to the calibration on yearly data, we now calibrate 12 localization radii instead of one, that is one for each calendar month. This of course takes about 12 times longer. Notably, the localization radii of the different months are rather similar: For steps of 250 kilometers we did not find radii differing by more than 500 km. In the future, we plan to leverage this similarity to speed up calibration after a radius is found for the first month.

The calibration for annual maximum temperatures with MESMER-X takes about 1.5 minutes for the small and 4 minutes for the large calibration dataset (Tab. 3). Again, the most time intensive step (about one third of the time for the large calibration dataset) is finding the localization radius, where in this use-case, the fist local minimum was found after 7 iterations for the small and after 14 iterations for the large dataset. The second most time-intensive operation is finding a suitable first guess, followed by fitting the conditional distribution coefficients (Tab. A3). The performance of the MESMER-X calibration depends heavily on how complicated the chosen expression for the conditional distribution is. Complex expressions, with many parameters and coefficients, will be slower than simple expressions. Besides, although the expressions are chosen based on physical and statistical reasons, the user may provide an ill-defined expression which will slow down the optimization algorithms or make them fail altogether. Finally, the first guess may still be not good enough despite the algorithm described in section 4.4. Here, we emulate local annual maximum temperature as samples of a normal distribution where the mean is linearly dependent on the global mean temperature and a variable scale parameter ($t_{max} \sim \mathcal{N}(\mu, \sigma)$, where $\mu = c_1 + c_2 \cdot t, \sigma = c_3$), which can be judged as a rather simple expression. While a generalized extreme value distribution would be a better choice for the distribution (Quilcaille et al., 2022), we use a normal distribution in this example to keep the number of parameters to fit smaller.

Furthermore, we provide time estimates for drawing emulations. To this end, we use the smoothed global mean temperature of the CMIP6 IPSL-C6MA-LR historical and SSP5-8.5 "r1i1p1f1" initial condition member as forcing data for MESMER and MESMER-X and the gridded temperature of the same member for MESMER-M, together with the calibrated parameters from the large ensemble calibration above. Drawing 100 samples of local annual mean temperature for 2652 gridcells and 251 time steps takes less than 20 seconds, while 1000 samples require approximately 2 minutes. Here, the emulation of local variability, i.e. sampling the AR(1) process is the most time intensive step (>50%, Tab. A1).





Drawing monthly samples for 100 realizations using the MESMER-M setup takes about 3 minutes. This is about 9 times slower than emulating annual mean temperatures. Note that one realization of monthly data has 12 times as many samples as yearly data. As a consequence, it is not possible to draw 1000 realizations of monthly temperature trajectories on a standard office machine, as it runs out of memory. Memory optimization is a goal of future MESMER developments but was not a priority for MESMER v1, because realizations can also be drawn sequentially by the user. This is also beneficial for having the emulations split into smaller files. Users should keep in mind however to set new seeds when drawing new realizations, otherwise one will end up with the same data.

Finally, Using the MESMER-X setup, drawing 100 samples of annual maximum temperature takes about one minute, while drawing 1000 samples takes about 4.5 minutes. For this setup, the most time intensive step is sampling the standard normal for few samples, while for the large case, it is transforming the standard normal samples to the desired target distribution (Tab. A3).

**Table 3.** Time needed for the calibration and emulation workflows of MESMER, MESMER-M and MESMER-X tested on CMIP6 data of IPSL-CM6A-LR with 2252 gridcells. Times were measured on one execution each, using the standard cProfiler of Python. All workflows where timed once, omitting data loading and saving, on a personal machine with an Apple M3 Pro Chip (Nov 2023) and 18 GB of memory.

| Component | Calibration | | Emulation | |
|---|---|---|---|---|
| | one member | ensemble | 100 emulations | 1000 emulations |
| | (251 samples) | (2380 samples) | | |
| **MESMER** | 25 s | 2 min 55 s | 13 s | 1 min 47 s |
| **MESMER-M** | 5 min 0 s | 25 min 26 s | 2 min 4 s | - |
| **MESMER-X** | 1 min 20 s | 3 min 42 s | 26 s | 4 min 30 s |

## 6 Summary and Outlook

Overall, the MESMER software package provides user friendly spatial climate model emulator functionalities needed to reproduce previous MESMER applications as well as providing a flexible basis for further MESMER developments. We restructured the code base into a modular, flexible, extensible and comprehensible software package, implemented a safe, modern and transparent data structure, and constructed extensive tests and documentation. Moreover, we showed that emulation of typical MESMER use-cases can be done in just minutes, while calibration of model can take between seconds (one member, annual mean temperature) and half an hour (multiple members and scenarios, monthly temperatures). The rewrite was not without its challenges: it was difficult to integrate and align the source code of the different components developed by multiple researchers over several years, some of whom were working at different institutions in the meantime. Much of the code was tailored to the specific use-cases of the experiments previously performed and integrating the code to be flexible for novel use-cases was challenging. Then, the numerical stabilization of the output required considerable time and research effort. Further, developments in external libraries were necessary to accommodate MESMER's use-case, foremost the recent inclusion of `DataTree`



in xarray. The resulting integrated code provides a much more versatile tool to the research community and will serve as basis
for follow-up MESMER developments.

The appropriate tools to meet the above-mentioned challenges were most importantly the implementation of unit and integration tests, which flag when functionalities are not behaving as expected. This did not only provide stability and functional testing of the code, but also enforced a deeper understanding of the functionalities on the developers side. Moreover, enforcing a four-eye principle of code review on a platform like github ensured that code was checked by someone else than the developer themselves before it was merged into the main code base. Lastly, version control using git allowed us to track changes in the code and trace issues to their root cause.

With these tools, the development team achieved a significant performance increase, not only with respect to the time needed for calibration and emulation, but also with respect to numerical stability. The first aspect, i.e. faster calibration and emulation, is essential for making emulators useful tools next to Earth System Models. The latter aspect, i.e. numerical stability, is important for reproducibility as well as reliability of the results. Moreover, completely rewriting research code with the goal to provide a comprehensive software package suitable for general and external use is rather rare and still under-recognized in academia (Trisovic et al., 2022; Reinecke et al., 2022; Nyenah et al., 2024, 2025). The present development of MESMER v1 should also be a motivation for more research code to become publicly available and suited for external use.

Future developments of MESMER should be twofold: (1) Scientific developments can focus on new applications of the existing capabilities of MESMER, such as attribution studies, exploration of counterfactual scenarios, overshoot scenarios; or new applications, such as including new variables or extending to multivariate emulations. (2) Technical developments of the code base should include further performance optimization of the fitting routines to make calibration even faster, especially of the conditional distributions, as well as memory optimizations for the emulation routines to enable drawing a larger number of emulations, notably for monthly temperatures. Immediate further developments include the integration of MESMER-M-TP into the code base.

In conclusion, the newly developed MESMER v1 modular Earth System Model emulator is a user-friendly open-source software that is available for the fast assessment of climate projections under various emission scenarios, as well as for numerous further applications of relevance for the climate research community and society.

*Code and data availability.* The pre-release version of the source code published for the initial submission of this manuscript is available on github (https://github.com/MESMER-group/mesmer/releases/tag/v1.0.0rc1) and zenodo (https://doi.org/10.5281/zenodo.17209019). The code used to produce the analyses in this paper are available on Zenodo (https://doi.org/10.5281/zenodo.17264436). CMIP6 data is available at https://esgf-node.llnl.gov/projects/esgf-llnl/. The reduced example data used for measuring performance improvements is part of the MESMER package. Pre-calibrated parameters for emulating annual mean surface temperatures are available to the community at https://doi.org/10.5281/zenodo.17250327.



*Author contributions.* Conceptualization: MH, VMB and LG. Formal Analysis and investigation: VMB. Funding acquisition: SIS. Methodology: VMB and MH. Software: VMB, MH and YQ. Visualization: VMB. Writing (original draft preparation): VMB, SS, and YQ. Writing (review and editing): MH, LG, SIS.

*Competing interests.* The authors declare no competing interests.

*Acknowledgements.* We would like to acknowledge the project EU SPARCCLE, (Socioeconomic Pathways, Adaptation and Resilience to Changing CLimate in Europe), EU Project Nr. 101081369, and the project SBFI Nr. 23.00376 of the Schweizer Staatssekretariat für Bildung, Forschung und Innovation, which enabled parts of this work. Moreover, we want to thank Lorenzo Pierini, Emmanuel Nyenah and Paul Eibensteiner for fruitful discussions and helpful inputs during the development phase of this work. We acknowledge the World Climate Research Program's Working Group on Coupled Modelling, which is responsible for the Coupled Model Intercomparison Project (CMIP),
and we thank the climate modeling groups, specifically for the IPSL-CM6A-LR model (Boucher et al., 2018, 2019) for producing and making their model output available. Furthermore, we are indebted to Urs Beyerle, Lukas Brunner, and Ruth Lorenz for pre-processing the CMIP6 data. Lastly, we thank Lea Beusch, Zeb Nicholls, Shruti Nath, Lorenzo Pierini, and Jonas Schwaab for their contributions to the MESMER code base.

**Appendix A**





**Table A1.** Time needed to calibrate and emulate annual mean temperature (MESMER workflow) for one ESM (IPSL-CM6A-LR). Calibration is once done only on one initial condition member of one scenario (r1i1p1f1 and SSP5-8.5) and once on multiple members of four scenarios (r1i1p1f1, r2i1p1f1, r3i1p1f1, r4i1p1f1 and SSP1-1.9, SSP1-2.6, SSP2-4.5, SSP3-7.0, and SSP5-8.5). The matching historical member is included once for every scenario member. This yields 251 samples for the one member calibration and 2380 samples for the ensemble calibration. Times for emulation are given for 100 and 1000 realizations. Times were measured on one execution each, using the standard cProfiler of python. All workflows where timed once, omitting data loading and saving, on a personal machine with an Apple M3 Pro Chip (Nov 2023) and 18 GB of memory.

| Task | Calibration one member (251 samples) | Calibration ensemble (2380 samples) | Emulation (100 realizations) | Emulation (1000 realizations) |
|---|---|---|---|---|
| fit Linear Regression | 0 min 0.0 s | 0 min 0.1 s | - | - |
| fit AR(1) process | 0 min 5.0 s | 1 min 0.4 s | - | - |
| find localized empirical covariance matrix | 0 min 17.1 s | 1 min 50.1 s | - | - |
| predict Linear Regression | - | - | 0 min 0.1 s | 0 min 3.1 s |
| draw tempo-spatially correlated samples | - | - | 0 min 12.1 s | 0 min 50.2 s |
| other | 0 min 3.3 s | 0 min 3.9 s | 0 min 0.4 s | 0 min 54.0 s |
| total | 0 min 25.4 s | 2 min 54.5 s | 0 min 12.6 s | 1 min 47.3 s |



**Table A2.** Time needed to calibrate and emulate monthly mean temperature (MESMER-M workflow) for one ESM (IPSL-CM6A-LR) on a standard office machine (). Calibration is once done only on one initial condition member of one scenario (r1i1p1f1 and SSP5-8.5) and once on multiple members of four scenarios (r1i1p1f1, r2i1p1f1, r3i1p1f1, r4i1p1f1 and SSP1-1.9, SSP1-2.6, SSP2-4.5, SSP3-7.0, and SSP5-8.5). The matching historical member is included once for every scenario member. This yields 251 samples for the one member calibration and 2380 samples for the ensemble calibration. Times for emulation are given for 100 realizations. No times can be given for drawing 1000 realizations as it is not possible to draw that many samples on a personal machine due to too small memory. Note that monthly emulations need 12 times the memory of annual emulations as in Tab. A1 and Tab. A3. Times were measured on one execution each, using the standard cProfiler of python. All workflows where timed once, omitting data loading and saving, on a personal machine with an Apple M3 Pro Chip (Nov 2023) and 18 GB of memory.

| Task | Calibration one member (251 samples) | Calibration ensemble (2380 samples) | Emulation (100 realizations) | Emulation (1000 realizations) |
|---|---|---|---|---|
| fit harmonic model | 0 min 14.1 s | 2 min 47.9 s | - | - |
| fit Yeo-Johnson Power transform | 1 min 22.0 s | 2 min 5.7 s | - | - |
| apply Yeo-Johnson Power transform | 0 min 16.4 s | 2 min 34.4 s | - | - |
| fit cyclo-stationary AR(1) process | 0 min 1.1 s | 0 min 5.2 s | - | - |
| find monthly localized empirical covariance matrices | 3 min 1.9 s | 16 min 35.7 s | - | - |
| predict harmonic model | - | - | 0 min 0.0 s | - |
| draw tempo-spatially correlated samples | - | - | 0 min 39.2 s | - |
| apply inverse Yeo-Johnson Power transform | - | - | 1 min 17.5 s | - |
| other | 0 min 3.9 s | 0 min 4.5 s | 0 min 6.8 s | - |
| total | 4 min 59.5 s | 24 min 13.3 s | 2 min 3.5 s | - |



**Table A3.** Time needed to calibrate and emulate annual maximum temperature (MESMER-X workflow) for one ESM (IPSL-CM6A-LR). Calibration is once done only on one initial condition member of one scenario (r1i1p1f1 and SSP5-8.5) and once on multiple members of four scenarios (r1i1p1f1, r2i1p1f1, r3i1p1f1, r4i1p1f1 and SSP1-1.9, SSP1-2.6, SSP2-4.5, SSP3-7.0, and SSP5-8.5). The matching historical member is included once for every scenario member. This yields 251 samples for the one member calibration and 2380 samples for the ensemble calibration. Times for emulation are given for 100 and 1000 realizations. The conditional distribution is based on a normal distribution with $\mathcal{N} \sim \mu = c_1 + c_2 \cdot T$, $\sigma = c_3$ where $T$ is annual global mean temperature. Times were measured on one execution each, using the standard cProfiler of python. All workflows where timed once, omitting data loading and saving, on a personal machine with an Apple M3 Pro Chip (Nov 2023) and 18 GB of memory.

| Task | Calibration one member (251 samples) | Calibration ensemble (2380 samples) | Emulation (100 realizations) | Emulation (1000 realizations) |
|---|---|---|---|---|
| find first guess | 0 min 32.0 s | 0 min 50.2 s | - | - |
| fit conditional distribution | 0 min 21.4 s | 0 min 29.7 s | - | - |
| apply Probability Integral Transform | 0 min 0.0 s | 0 min 0.3 s | 0 min 4.6 s | 3 min 17.5 s |
| fit AR(1) process | 0 min 5.0 s | 0 min 58.2 s | - | - |
| find localized empirical covariance matrix | 0 min 17.4 s | 1 min 19.5 s | - | - |
| draw tempo-spatially correlated samples | - | - | 0 min 21.3 s | 1 min 12.4 s |
| other | 0 min 4.2 s | 0 min 4.0 s | 0 min 0.0 s | 0 min 0.0 s |
| total | 1 min 20.1 s | 3 min 41.8 s | 0 min 25.9 s | 4 min 29.8 s |



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
