# Peer review of "MESMER v1.0.0: Consolidating the Modular Earth System Model Emulator into a Sustainable Research Software Package"

_EGUsphere, 2025_

## Referee Comment (RC1)

This paper presents the modular climate model emulator (MESMER v1), its components, and strategy for integration, and serves as detailed documentation in addition to the online documentation for users. The authors also demonstrate that this rewritten emulator now meets sustainable research software standards. The output of the emulator includes annual and monthly mean temperature and several climate extreme indicators, such as maximum annual temperature.

The paper provides a thorough description of the emulator, improvements made as well as performance and sustainability assessment. However, prior to publication, the following comments and questions should be addressed:

**General comments:**

*Motivation of study:*

The motivation of this study appears to focus mainly on making the emulator accessible and providing documentation on how to use it (as inferred from the introduction: Lines 30-31). However, users may also want to be able to read and understand the model source code, extend and modify its functionality, and contribute to maintenance (if possible, e.g., by reporting bugs) in order to improve the overall software sustainability of the emulator. Therefore, I recommend revising the motivation around lines 30–31 to more fully expand on these aspects.

*State of old MESMER software:*

What is currently not clear to me is the state of the old (legacy) MESMER software. You only hint at this with statements such as *organic growth of the code base* (Section 1: Introduction, lines 36–38) and *development of software by multiple researchers without a focus on software design* (Section 3: Design of MESMER v1, lines 98–99). Was documentation available for the old software? Was the old software easily accessible by users? Is the code of the legacy MESMER difficult to read and understand? What is the state of code comments in the old code? Were the separate components of old MESMER all integrated as shown in this paper? Did the old software have a well-defined software architecture? I expected to see a section that answers and expands on these points.

This would also be beneficial during your assessment (Section 3.6: Assessment of the redesigned code base), as it would help to understand how much the rewritten code improves upon the old one. Additionally, did the state of the old code require rewriting everything from scratch (or only parts of it)? This also highlights the importance of making

software sustainable, as complete rewriting requires significantly more time and effort. This is an issue that is now slowly gaining attention in academia.

*Performance assessment:*

Performance assessment (Section 5) was mainly focused on runtime. It would be helpful for the paper to also include some visualized plots showing the reproduction of old results or even improvements over previous results. For example, in Section 4.3, *Switch to a cyclo-stationary AR(1) process for monthly emulations*, you state that the cyclo-stationary AR(1) represents variability better than the AR(1) process. However, the corresponding plot is not shown.

**Specific comments**

*Section 1: Introduction*

Line 15: Please provide citation(s)

*Section 3: Design of MESMER v1*

Line 102: Please check the citation. Should it be Anzt et al., 2021?

Lines 112-113: Is mesmer.proba the same as mesmer.distrib in Figure 2? If so, could you please make it consistent?

Lines118-119: For data handling functionality, I don't seem to see mesmer.anomaly.calc_anomaly in Fig. 2. (mesmer),

Lines 121-124: Making this workflow for calibration and emulation creation readily available would already benefit users and improve reproducibility. Is this workflow already available?

Lines 154-156: Please provide citation.

Line 206: Should this be 238 instead of 269 as in Table 2?

Line 276-278: "... We were able to fix these issues by switching to more precise routines", This line is unclear to me. Which routines were used to address the numerical stability problem?

*Section 5: Performance assessment*

There are several instances where runtimes are mentioned as referring to Table 3, but Table 3 is not actually referenced. For example, in Lines 367–368 and 390–391. Please revise these references to make them clear

Line 355: Should this be 3 minutes rather than 4 minutes?

Line 400: Should this be 30 seconds rather than 1 minute?

*Other questions:*

Is there a containerized solution for the new MESMER, such as a Docker container, to enable running it on different computers and facilitate reproducibility?